# Local versus farfield control on South Pacific Subantarctic mode water variability

Ciara Pimm[1,4], Andrew J. S. Meijers[2], Dani C. Jones[2,3], and Richard G. Williams[1]

[1]Department of Earth, Ocean and Ecological Sciences, School of Environmental Sciences, University of Liverpool, Liverpool, UK
[2]British Antarctic Survey, NERC, UKRI, Cambridge, UK
[3]Cooperative Institute for Great Lakes Research (CIGLR), University of Michigan, Ann Arbor, MI, USA
[4]Woods Hole Oceanographic Institution, Woods Hole, MA, USA

**Correspondence:** Ciara Pimm (c.pimm@liverpool.ac.uk/ciara.pimm@whoi.edu)

**Abstract.** In the South Pacific Subantarctic mode water (SAMW) formation region, central and eastern pools of SAMW have been found to be linked to winter mixed-layer thicknesses that vary strongly interannually and out of phase across the basin. This mixed-layer variability is associated with peaks in sea level pressure variability at a quasi-stationary anomaly situated between the two pools. To investigate how surface forcing, as well as the propagation of upstream anomalies, affects the formation of these SAMW pools, a set of adjoint sensitivity experiments with a density-following feature are conducted. Adjoint sensitivities reveal local cooling can lead to an increase in SAMW pool volume through mixed-layer depth changes and the lateral movement of the northern boundary of the pool. In addition, upstream warming along the Antarctic Circumpolar Current can lead to an increase in SAMW pool volume through lateral density surface movement shifting the southern boundary polewards. The density properties are advected from upstream to the downstream pool over one year. Optimal conditions for SAMW formation involve a combination of local cooling and upstream warming of SAMW formation sites. Hence, South Pacific SAMW variability is particularly sensitive to atmospheric modes which lead to a dipole in heating across the formation sites.

## 1 Introduction

Pools of Subantarctic mode water in the South Pacific Ocean are climatically important due to their role in the uptake and transport of heat, carbon, and nutrients around the global oceans (Williams et al., 2023). The overturning circulation in the Southern Ocean has a two-cell structure (Marshall and Speer, 2012). The upper cell involves the ventilation of circumpolar deep water (CDW) to form northward flowing mode and intermediate waters. The deeper cell is formed via upwelling CDW being transformed through brine rejection during ice formation. The overturning circulation is a combination of a time-mean circulation and a time-varying eddy circulation.

Subantarctic mode water (SAMW) is a subsurface water mass that can be identified by its low absolute value of potential vorticity (McCartney, 1977), where a water mass with low potential vorticity is a vertically well mixed water mass. SAMW is formed in areas of deep winter mixed layers through surface ventilation and winter heat loss on the northern side of the Antarctic Circumpolar Current (ACC) (McCartney, 1982). The springtime shoaling of mixed layer depths leaves behind a signal of well mixed water with low potential vorticity, known as SAMW, and is subsequently advected away from the formation sites. If not subsequently obducted in later winters, SAMW circulates into the subtropics where it may remain for many decades (Jones et al., 2016). SAMW is found in the Southern Ocean in all three basins to the north of the ACC, with the Pacific Subantarctic mode waters being the densest and the Atlantic being the lightest (Sallée et al., 2010). SAMW is important as it is the main pathway of heat and carbon uptake from the Southern Ocean into the global ocean. The Southern Ocean dominates global ocean heat uptake (Williams et al., 2023) and contributes 40% of $CO_2$ uptake historically (Frölicher et al., 2015). SAMW has been found to be thickening (Gao et al., 2018), which is contributing to the bulk of recent ocean warming (Li et al., 2023). As the thickness of Subantarctic mode waters is seen to be very important, density coordinates are a better fit than depth coordinates for assessing changes in the formation rates and properties of mode waters.

Due to the advent of the Argo time series there are now orders of magnitude more ocean observations available than at the turn of the century, particularly in the Southern Ocean (Johnson et al., 2022). An Argo time series allows for interannual and internal variability in SAMW regions to be considered (Li et al., 2021). In the Pacific and Indian Subantarctic mode water formation regions, central and eastern pools have been found to have wintertime thicknesses that vary strongly interannually and out of phase across each basin (Figure 1, Figure S1) (Meijers et al., 2019). This thickness variability is associated with changes in atmospheric forcing, as evident in peaks of variability in sea level pressure between the central and eastern pools in each basin. Therefore, each basin wide SAMW pool can be further split into a central and eastern pool (Cerovečki and Meijers, 2021). Cerovečki and Meijers (2021) found that these dipoles in thickness of the pools within each ocean (Indian and Pacific) basin were driven by surface heat flux differences on either side of the basin, over each pool, and that these differences arose due to sea level pressure anomalies situated between the pools. For positive (negative) anomalies, warm (cool) air would be drawn from the north (south) over the central SAMW pools to the west of the sea level pressure anomaly, driving anomalous heat gain (loss) and resulting in shoaling (deeper) mixed layers and smaller (increased) SAMW layer volumes. The opposite would happen over the pool to the east of the sea level pressure anomaly, resulting in a clear mixed layer and SAMW dipole across the basin. Furthermore, Cerovečki and Meijers (2021) found that these sea level pressure anomalies tend to be correlated across the Pacific and Indian basins, as part of the wider wave number 3 mode of pan-Antarctic sea level pressure variability. However, it was also found that this process was not always the case, particularly in the presence of strong ENSO events, which primarily only affected the Southeast Pacific basin. Meijers et al. (2019) found that the large scale atmospheric modes, SAM and ENSO, mapped onto the sea level pressure anomalies and had similar fingerprints on the mixed layer depth patterns. This similarity suggests that SAM and ENSO anomalies may reinforce or weaken the dipole patterns depending on their relative phases. For example, positive ENSO is associated with heat gain over the central box and heat loss over the eastern box, so providing a west-east dipole in the thermal forcing. While these studies primarily focused on the role of the local heat fluxes in driving the SAMW formation site mixed layer depth variability, both Cerovečki and Meijers (2021) and Meijers et al. (2019),

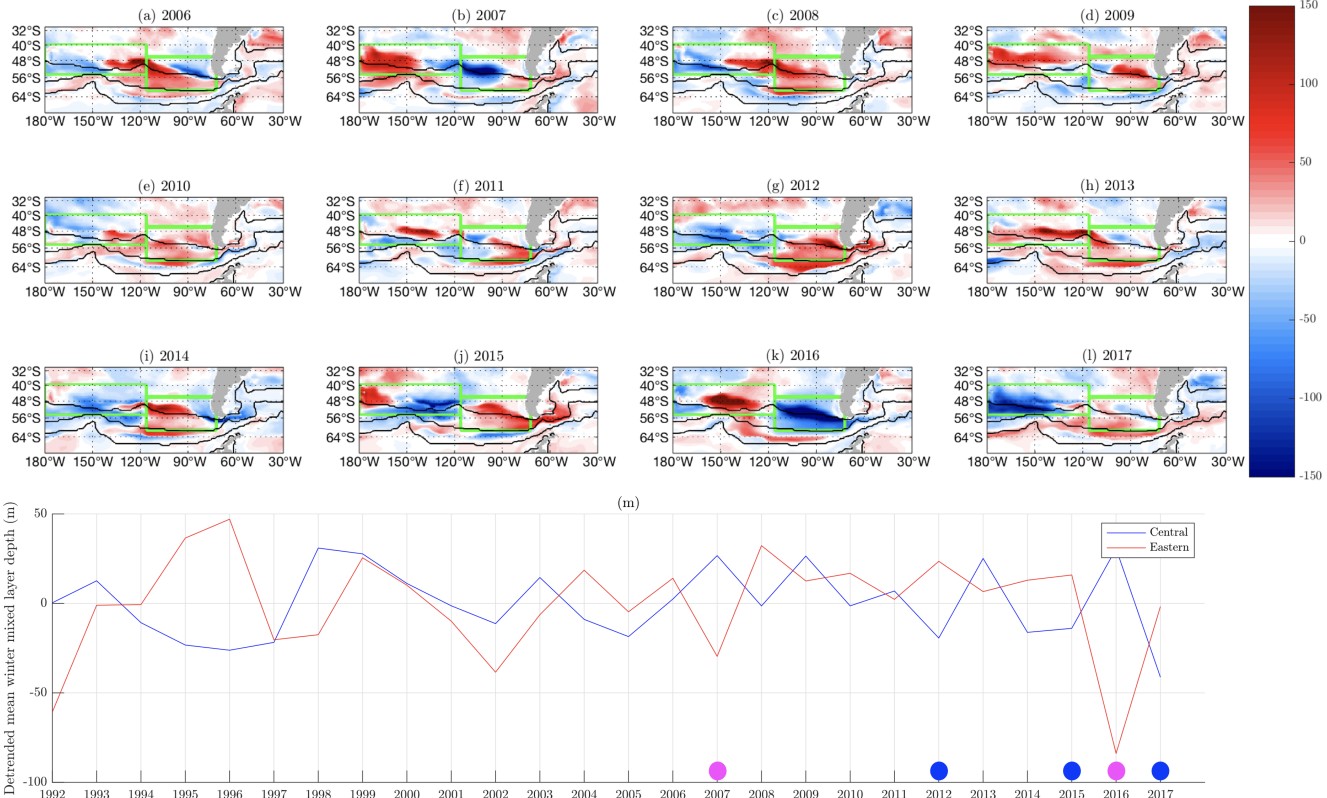

**Figure 1.** Winter mixed layer depth anomalies (m) for years (a) 2006 - (l) 2017 from ECCOv4r4 data. Winter mixed layer depth is calculated using the depths where neutral density increase from the surface is $\geq 0.03$ kg m$^{-3}$. Green boxes show the central and eastern Subantarctic mode water formation pools. Black contours show sea surface height proxies(0.25, 1.0, 1.5 m) for the fronts of the ACC. (m) Time series from 1992 – 2017 of detrended winter mixed layer depth (m) within the central (blue) and eastern (red) boxes. Dots on the x-axis show years with strong differences in thickness between the central and eastern pools, magenta dots have positive central and negative eastern anomalies, whilst blue dots have negative central and positive eastern anomalies.

noted the presence of a clear one year lag between mixed layer depth properties transferring from the central Pacific pool to the eastern Pacific pool. Both studies suggest, but do not quantify, the potential for advective processes to set, or precondition, the response of the downstream SAMW pools.

The main aims of this study are to understand the effects of different surface forcing on Subantarctic mode water pool volumes. It is known that heat fluxes and wind stresses are the most important in driving Subantarctic mode water volume change, however the local verses farfield contributions have not been quantified, despite several studies indicating that advection of upstream water masses, and their modification, is likely to be important in setting SAMW mixed layer and subducted properties. To provide this quantification, adjoint sensitivity experiments are used to quantify when and where each forcing affects each Subantarctic mode water pool over the South Pacific.

Adjoint sensitivity analysis is a technique used in meteorology and oceanography to efficiently estimate the sensitivity of numerical model outputs with respect to various inputs (Errico, 1997). This process involves starting from a quantity of interest, known as the objective function, and propagating its sensitivities backward through a linearised version of the model. This allows for efficient determination of how various model variables affect the objective function, facilitating applications in data assimilation, parameter estimation, stability analysis, and sensitivity studies.

The concept of adjoint models began to gain traction in the meteorological community in the late twentieth century, as evidenced by literature reviews produced around that time (Courtier et al., 1993). In the mid-1990s, building on the foundation established in meteorology, adjoint models were applied to estimate ocean circulation using data assimilation (Marotzke and Wunsch, 1993). Although many of the ocean adjoint modelling tools at this time were developed with state estimation in mind, they also proved useful for adjoint sensitivity experiments, such as in a study of the sensitivity of Atlantic heat transport

(Marotzke et al., 1999; Bugnion et al., 2006). Building on this foundational work, adjoint sensitivity studies have been carried out on many aspects of Atlantic overturning (Köhl, 2005; Czeschel et al., 2010, 2012; Sévellec and Fedorov, 2016), such as heat and volume transport of various limbs of this circulation system (Heimbach et al., 2011; Kostov et al., 2019). Adjoint sensitivity experiments can also be useful for studying smaller-scale current systems, such as coastal currents (Moore et al., 2009; Verdy et al., 2014), throughflows (Nguyen et al., 2020), and tide surge modelling (Wilson et al., 2013; Warder et al.,

2021). In addition to transport, it has also been applied to quantify the sensitivity of integral quantities such as volume-and-time mean heat content to local and remote atmospheric forcing (Jones et al., 2018b, 2020b; Boland et al., 2021b; Meijers et al., 2023b). Notably, this method can also be used for dynamical attribution and uncertainty analysis, allowing one to quantify the contribution of various forcing factors to the variability of the quantity of interest (Pillar et al., 2016, 2018; Smith and Heimbach, 2019; Stephenson and Sévellec, 2021; Kostov et al., 2021) or to identify how to optimally excite variability on a

selected timescale (Zanna et al., 2011, 2012; Sévellec and Fedorov, 2015). More recently, the method has been applied to the sensitivity of volume (Kostov et al., 2024) and stratification (Pimm et al., 2024b). If the objective function is defined as the integral of a tracer concentration, the sensitivity fields act as an adjoint tracer, revealing the source waters for the control volume (Fukumori et al., 2004; Chhak and Di Lorenzo, 2007; Song et al., 2016). It has been used for biogeochemical properties, such as productivity, air-sea carbon flux, and carbon sequestration efficiency (Hill et al., 2004; Dutkiewicz et al., 2006). It has also

been used to diagnose the causes of bottom pressure variations, leading to insights on how atmospheric variability affects large-scale circulation (Fukumori et al., 2007b, 2015). The tool has also been used in glaciology to quantify the sensitivity of the melt rate and volume (Heimbach and Bugnion, 2009; Goldberg et al., 2020). More recently, adjoint sensitivity experiments have been used to inform observing system design by quantifying the proxy potential of selected observing sites (Loose et al., 2020; Loose and Heimbach, 2021). The wide range of adjoint sensitivity applications highlights its versatility.

Firstly, the ECCOv4r4 model and adjoint approach are outlined, Section 2.1. Preexisting adjoint machinery uses depth coordinates (Boland et al., 2021a; Jones et al., 2020a), however this does not take into account mode water thickness changes. To address the aims a new density-following adjoint model feature is created, which is discussed in Section 2.2, which allows the variability of volume of Subantarctic mode water in the central and eastern South Pacific to be considered more explicitly. In Section 2.3, the experimental design is discussed which sets out the objective functions used and the horizontal and vertical

areas chosen for each South Pacific Subantarctic mode water pool experiment. Section 3 considers the resulting impacts, which are defined as the adjoint sensitivities convolved with forcing variability, to understand when and where different surface forcings impact each Subantarctic mode water pool separately. Then, the linear causal mechanisms implied by the adjoint sensitivities are tested using forward perturbation experiments, Section 4. Finally, the wider context of the work is set out in the discussion and conclusions.

## 2   Experimental Design

### 2.1   The ECCOv4r4 State Estimate and the Adjoint Approach

Numerical modelling experiments are used to understand the local and farfield influences of surface forcing, including heat and freshwater fluxes, and wind stresses, on the central and eastern Southeast Pacific Subantarctic mode water pools. Forward and adjoint modelling is completed using the ECCOv4r4 (Estimating the Circulation and Climate of the Ocean version 4 release 4) state estimate (ECCO Consortium et al., 2020). ECCOv4r4, which covers a 26 year time period from 1992 to 2017, is used for all numerical experiments completed in this study. ECCOv4r4 is a physically-consistent ocean state estimate, which combines ocean observations with the MITgcm model equations to give a more accurate description of the oceanic system with insight into the underlying processes (Forget et al., 2015a). Furthermore, the ECCOv4r4 state estimate has been shown to have good agreement with observations in studies of Southern Ocean mode water properties (Jones et al., 2018a; Boland et al., 2021a; Forget et al., 2015b).

In a more traditional modelling approach, forward perturbation experiments define a quantity of interest and chosen area, which is then perturbed by a small amount. The effects of the perturbation on the model state are seen in difference from the unperturbed control state. For adjoint sensitivity experiments one quantity of interest, or objective function, is chosen and then the adjoint sensitivities of this function to every independent variable at each time, latitude, longitude, and depth are calculated backwards in time (Jones et al., 2018a; Pimm et al., 2024a). The resulting adjoint sensitivities are linear gradients of the objective function to all model independent variables at all positions and times. These adjoint sensitivities are used to identify potential causal mechanisms affecting ocean properties. However, a limitation of adjoint sensitivity experiments is that the adjoint sensitivities produced are only linear approximations. Previous work has suggested that the linear approximation is useful for several years to decades when using large scale quantities of interest (Jones et al., 2018a; Boland et al., 2021a; Pimm et al., 2024a).

### 2.2   Density-following Feature

Previously within MITgcm a fixed vertical mask, defined by depth levels constant in time, has been used when employing adjoint sensitivity experiments. The fixed vertical levels lead to the objective function being integrated over a fixed volume. However, for examining mode waters this approach can be limited as mode waters span across multiple depths over time and are physically defined by density or potential vorticity characteristics, which shift with time. This depth choice leads to adjoint

sensitivity experiments not being able to capture the full mode water features properly, as they do not capture the isopycnal thickness changes that are a defining characteristic of SAMW variability. To address this limitation, a density-following feature (Appendix A) is added into MITgcm that allows the objective function to be defined as an integral between two neutral density surfaces that move in time (Campin et al., 2022). A problem that this new feature resolves is that mode water sensitivities previously had to be interpreted as information for the specific mode water pool defined combined with other water masses (Meijers et al., 2023a). However, the new density-following feature can be used to isolate specific water masses only, as defined by their density structure. Therefore, this feature will provide a more accurate interpretation of water masses from the adjoint sensitivities.

This addition of the objective function integrated over density surfaces that move vertically in time is achieved using a sigmoid function. The density-following feature is used to update the vertical mask at each time step and then the linear partial differentials to this objective function are calculated in the adjoint step of the model run.

In addition, for the density-following feature, another feature is added to MITgcm to permit an objective function of volume between the upper and lower bounding density surfaces to any model independent variables (Campin et al., 2023). The objective function of volume between the two surfaces is given as:

$$J = \frac{1}{\Delta t} \int\limits_{t_1}^{t_2} \int\limits_{\sigma_l}^{\sigma_u} \int\limits_{A} \, dA \, d\sigma \, dt, \tag{1}$$

where $J$ is the objective function, $\Delta t$ is the time interval between $t_1$ and $t_2$ over which the objective function is applied, $\sigma_l$ and $\sigma_u$ are the lower and upper bounding density surfaces, and $A$ is the area over the horizontal area chosen.

After both features were created, testing was conducted using both simplified models and the full ECCOv4r4 state estimate. After these tests were completed successfully, the density-following feature (Campin et al., 2022) and the volume objective function feature (Campin et al., 2023) were added to MITgcm using GitHub.

## 2.3 Defining the Central and Eastern Pools

In the South Pacific Subantarctic mode water formation region, central and eastern Subantarctic mode water pools have been found to have wintertime thicknesses that vary strongly interannually and out of phase across the basin (Meijers et al., 2019). The South Pacific Subantarctic mode water formation region is chosen here due to it being the focus of several previous studies (Cerovečki and Meijers, 2021; Meijers et al., 2019; Li et al., 2021), and also is uniquely subject to ENSO forcing. The South Indian SAMW formation region is also found to have central and eastern Subantarctic mode water pools which vary in thickness, so results found here may be more widely applicable.

The central and eastern regions are identified using local maximum winter mixed layer depth and low absolute potential vorticity from ECCOv4r4 (Figure 1), which are separated by the Pacific Antarctic Ridge longitudinally. Meijers et al. (2019) found that the wintertime thickness of the central and eastern pools of Subantarctic mode water formation vary out of phase with each other. Winter mixed layer depth anomalies have opposing signs in each pool which alternate each year (Figures 1 (a) - (l)). The relative phases of the dominant atmospheric modes in the Southern Ocean, SAM and ENSO, drive the variability of SAMW

volume and heat content. Expanding on this research, Cerovečki and Meijers (2021) show that the local sea level pressure anomalies are the main drivers of the variability in SAMW volume, although SAM and ENSO map onto these anomalies. When

SAM and ENSO are in phase with each other, positive enhanced westerly winds and associated cool Ekman transport lead to a deepening of the more southern eastern pool which is reinforced by enhanced southerlies. When the atmospheric modes are out of phase there is a less coherent response in the Subantarctic mode water formation pools, meaning the interannual dipole signal is stronger in some years than others, and these years are identified by magenta and blue dots along the time winter mixed layer depth time series (Figures 1 (m)). The central and eastern pool variability in South Pacific SAMW is generally

only seen in the later half of the ECCOv4r4 state estimate. Furthermore, towards the start of the ECCOv4r4 state estimate it is not as reliable due to lack of observations in the Southern Ocean. Characteristic density surface outcrop positioning is further to the north if the Subantarctic mode water pool has a positive winter mixed layer depth anomaly and further to the south if the Subantarctic mode water pool has a negative winter mixed layer depth anomaly (Figures 2 (a), (b)). Over the seasonal cycle of the mixed layer, the outcrop positions of the density surfaces moves north in winter and south in summer.

This information, along with Figures S2 and S3, is used to define the horizontal and density masks that are used to create the objective functions for the following adjoint sensitivity experiments. The objective function of volume in the central and eastern Subantarctic mode water pool respectively is given by:

$$
J_C(V) = \frac{1}{\Delta t} \int_{t_1}^{t_2} \int_{\sigma_{lC}}^{\sigma_{uC}} \int_{A_C} dA d\sigma dt, \tag{2}
$$

$$
J_E(V) = \frac{1}{\Delta t} \int_{t_1}^{t_2} \int_{\sigma_{lE}}^{\sigma_{uE}} \int_{A_E} dA d\sigma dt, \tag{3}
$$

where $\Delta t$ is the time interval between $t_1$ and $t_2$ which spans from January 2017 to December 2017; $\sigma_{uC}$, $\sigma_{lC}$, $\sigma_{uE}$, $\sigma_{lE}$ are the upper and lower bounding density surfaces for each of the central and eastern pools; and $A_C$, $A_E$ are the latitude-longitude areas of each Subantarctic mode water pool. The bounding neutral density surfaces are given by:

   – $\sigma_{uC} = 26.7$ kg m$^{-3}$, $\sigma_{lC} = 27.1$ kg m$^{-3}$,

– $\sigma_{uE} = 26.9$ kg m$^{-3}$, $\sigma_{lE} = 27.15$ kg m$^{-3}$.

The horizontal area of each pool is defined by:

   – $A_C = 180° − 116°$ W, $40° − 55.3°$ S,

   – $A_E = 116° − 72°$ W, $47° − 62°$ S.

An experiment for each Southeast Pacific Subantarctic mode water pool is then run individually on ARCHER2 to give the

outputs of the adjoint sensitivity of volume to each independent variable.

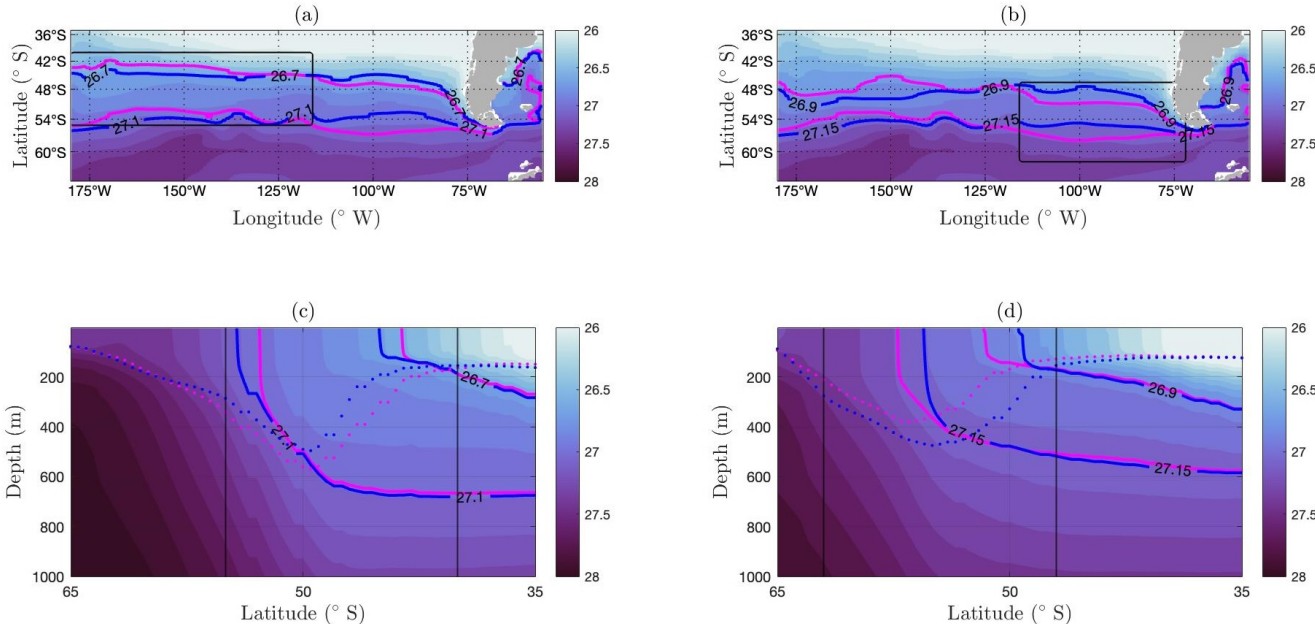

**Figure 2.** (a) Time mean neutral density at the surface over longitude and latitude shown for central region density surfaces (26.7 and 27.1 kg m$^{-3}$). Blue contours are for when the central pool mixed layer depth anomaly is in negative phase and the eastern pool mixed layer depth anomaly is in positive phase. Magenta contours are when central pool mixed layer depth anomaly is in positive phase and the eastern pool mixed layer depth anomaly is in negative phase. Black box shows the Central Pacific pool. (b) Time mean neutral density at the surface over longitude and latitude shown for eastern region density surfaces (26.9 and 27.15 kg m$^{-3}$). Black box shows the Eastern Pacific pool. (c) Time mean neutral density over latitude and depth averaged over the longitudes of central region. Blue dotted line is mean mixed layer depth over central longitudes when central pool mixed layer depth anomaly is in negative phase and the eastern pool mixed layer depth anomaly is in positive phase. Magenta dotted line is mean mixed layer over central longitudes depth when central pool mixed layer depth anomaly is in positive phase and the eastern pool mixed layer depth anomaly is in negative phase. Black lines show the Central Pacific pool latitudes. (d) Time mean neutral density over latitude and depth averaged over the longitudes of eastern region. Black lines show the Eastern Pacific pool latitudes.

## 3 Impacts

Impacts, defined as the product of adjoint sensitivities and the variation in the input variable, are the focus of this study. For example, the surface heat flux impacts on volume, $I_{\mathcal{H}}$, are calculated from the product of the adjoint sensitivities of volume to heat flux and the heat flux anomalies:

$$I_{\mathcal{H}} = \frac{\partial V}{\partial \mathcal{H}} \times (\mathcal{H} - \overline{\mathcal{H}}), \tag{4}$$

where $\mathcal{H}$ represents surface heat flux, $\partial V / \partial \mathcal{H}$ represents the sensitivity of volume to heat flux, and $(\mathcal{H} - \overline{\mathcal{H}})$ represents surface heat flux anomaly relative to the multi-year mean annual average, which is calculated including the seasonal cycle. In ECCOv4r4, a positive surface heat flux represents a heat loss from the ocean and a negative surface heat flux represents a heat gain by the ocean. The impacts for freshwater flux, eastward wind stress, and northward wind stress are calculated in the same way using their respective variables. An advantage of using impacts over adjoint sensitivities is that the influence of different surface forcings can be more easily compared to each other, due to being of the same unit, and as they are more 'realistic' in terms of relative magnitude. Both adjoint sensitivities, may be seen in Figures S4 and S5, and impacts, may be seen in Figures S6 and S7, are useful to understand processes and provide complementary information as to the intrinsic sensitivity of volume and the extent that surface forcing affects volume.

### 3.1 Quantifying the relative importance of surface forcing

To determine the relative importance of the upstream versus local impacts on each control volume, the impacts are separated by region based on whether the impacts are within the adjoint control volume or anywhere else in the global ocean, referred to as local and nonlocal respectively, such that:

$$I_{total} = I_{local} + I_{nonlocal}, \tag{5}$$

for each different forcing variable considered. Separating the impacts into local and nonlocal components helps to quantify which regions are most important for each surface forcing (Figure 3). For Figure 3, both $I_{local}$, $I_{nonlocal}$ are integrated over their respective domains, to give a time series showing the average evolution of the impact over time for each region. Total impacts, $I_{total}$, may be seen in Figure S8 in order to better compare the relative magnitude of each separate impact.

Positive heat flux impacts represent areas where either cooling leads to an increase in Subantarctic mode water volume or warming leads to a decrease in Subantarctic mode water volume. Negative heat flux impacts represent areas where either cooling leads to a decrease in Subantarctic mode water volume or warming leads to an increase in Subantarctic mode water volume. These signs carry over for the other independent variables, freshwater flux and wind stresses. The size of the volume change that can be inferred from the heat flux impacts can be considered over one grid cell. Impacts are given per grid cell for a snapshot of a 14-day period with units of the objective function per units of the independent variable. It is assumed that a snapshot of a 14-day period is similar to the average of the 14-day period. In order to determine the volume change that is ultimately caused by a change in surface forcing at a specific lead time, impacts are considered at each grid point. The net effect is found by integrating over all grid cells and lead times.

Surface heat flux has the largest impact on Subantarctic mode water volume in the central Pacific region initially (Figure 3 (a)). Preceding winter (JAS), local heat flux is most important, with the farfield, or upstream, contribution only providing a small impact (Figure 3 (a)). Surface heat flux is also the most important impact on Subantarctic mode water volume in the eastern Pacific region (Figure 3 (e)). In contrast to the central pool, for the eastern pool the nonlocal contribution becomes very important after one year lead time. At 1.5 years lead time the most important impact on the eastern Subantarctic mode water pool is upstream heat flux, which is positioned within the central Subantarctic mode water pool (Figure 4 (e)). At 1.5 years lead time, the central pool has local impacts of size approximately $+5 \times 10^{12}$, whilst the nonlocal impacts are less than $\mathcal{O}(10^{12})$. Compared to the interannual and seasonal variability of the volume of the central mode water pool, the impacts are of the same order of magnitude, showing that heat flux is very important in the formation of the mode water pool. By contrast, at 1.5 years lead time the eastern pool has local impacts less than $\mathcal{O}(10^{12})$ but has nonlocal impacts of size approximately $-2 \times 10^{12}$. The negative sign of the nonlocal impacts indicates, that the central region is where warming can lead to an increase in volume in the eastern pool; the underlying mechanisms of this process are explored in Section 4.2. The interannual and seasonal variability of the volume of the eastern mode water pool is of $\mathcal{O}(10^{12})$, showing that other impacts are also important in the formation of the eastern pool. This analysis helps to quantify the previously suggested role of advective processes in preconditioning the layer thickness of the downstream eastern Subantarctic mode water pool (Cerovečki and Meijers, 2021; Meijers et al., 2019).

Eastward and northward wind stresses also play a role on lead times of 3 years or more (Figure 3 (b), (c)). Local and nonlocal eastward wind stress are both equally important in impacting central Pacific Subantarctic mode water volume up to 3 years lead time when nonlocal eastward wind stress dominates (Figure 3 (b)). At approximately 3 years lead time nonlocal eastward stress also dominates over the surface heat fluxes. Local and nonlocal northward wind stress plays a small role in impacting the volume of the central Subantarctic mode water pool (Figure 3 (c)). Similarly, the eastward wind stress impacts are larger than the northward wind stress impacts for the eastern Subantarctic mode water pool (Figures 3 (f), (g)). After 3 years lead time the total eastward wind stress is a more dominant contribution over nonlocal heat fluxes for the eastern Subantarctic mode water pool (Figure 3 (f)). Freshwater flux has a very small impact on the volume of both the central and eastern Subantarctic mode water pools (Figures 3 (d), (h)).

Initially, local heat flux is the dominant impact on volume, as seen by the amplitude of its impacts. On lead times longer than 1.5 years, the central pool continues to be dominated by local heat fluxes, but the eastern pool is dominated by upstream heat fluxes. This shows the importance of upstream heat fluxes in the eastern Subantarctic mode water pool due to advection from the upstream central pool to downstream eastern pool via the mean flow. The fact that no such nonlocal dominance occurs within the central pool indicates the relative dynamic isolation of the Pacific from the Indian formation sites (Cerovečki and Meijers, 2021). In contrast, the volume of the eastern pool is strongly impacted by the propagation of previously formed SAMW and mixed layers from the central Pacific in the previous winter. This 1.5 year lead time aligns with the timescales hypothesised by Meijers et al. (2019), and represents the first quantification of this process. After 3 years lead time, local and nonlocal eastward wind stresses play an important role in impacting the volume of both Subantarctic mode water pools.

During the first year lead time in the Central region, the maximum overall impact is made up of contributions of 88% from surface heat flux, 6% from eastward wind stress, and 5% from northward wind stress. Of this total surface heat flux forcing for

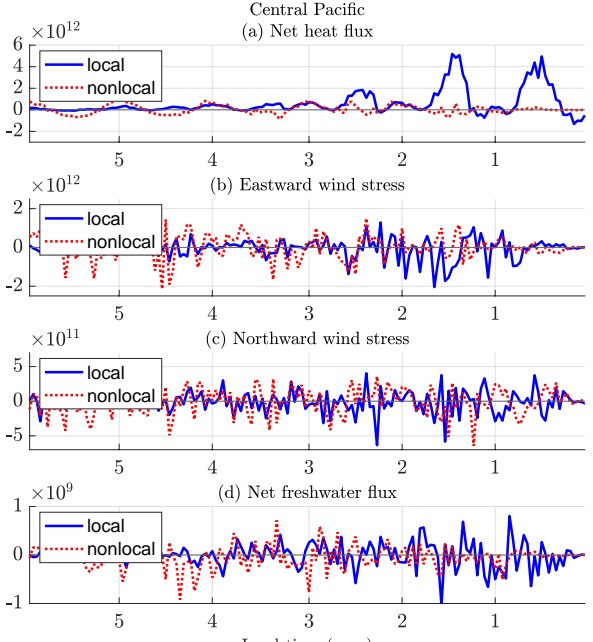
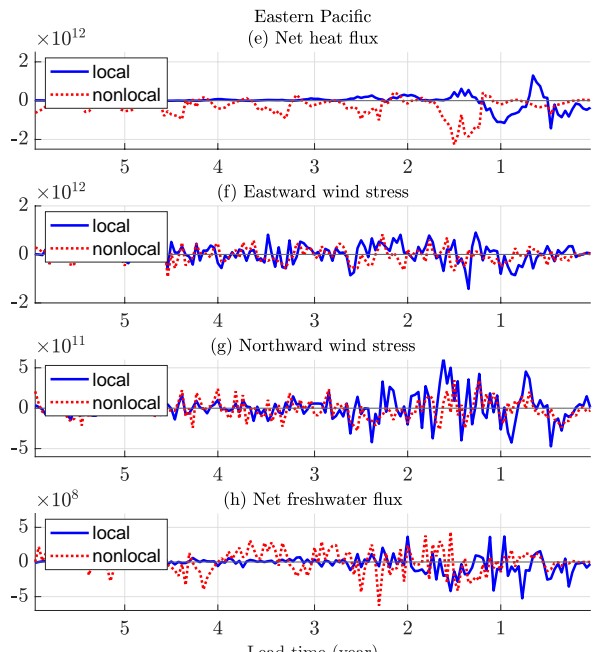

**Figure 3.** (a) Impacts of surface heat flux for the central pool at the surface (anomalies multiplied by sensitivities, m$^3$) split into local and nonlocal contributions. (b) Same for eastward wind stress (m$^3$). (c) Same for northward wind stress (m$^3$). (d) Same for surface freshwater flux (m$^3$). (e – h) Same for the eastern pool. Note y-axis for each separate impact is different.

the Central region, 94% is from local sources and 5% is from nonlocal sources. During the first year lead time in the Eastern region, the maximum impact is made up of contributions of 43% from surface heat flux, 43% from eastward wind stress, and 13% from northward wind stress. Of this total surface heat flux forcing for the Eastern pool 85% is from local sources and 14% is from nonlocal sources. Of this total eastward wind stress forcing for the Eastern pool 61% is from local sources and 38% is from nonlocal sources. Hence, the impact is most strongly affected by local forcing and the contribution of nonlocal forcing varies with the type of forcing and the region.

## 3.2 Effect of surface heat flux on volume of waters within density outcrops

As surface heat flux is revealed to be the dominant impact on the volumes of both the central and eastern Subantarctic mode water pools, the present subsection focuses on the role of the surface heat flux to reveal the underlying mechanism.

Latitude-longitude, wintertime snapshots of heat flux impacts are considered at the surface for each Subantarctic mode water pool (Figure 4). Figure 4 compares the heat flux impacts with the position of the bounding density surface outcrops. Initially, the impacts are mainly positive along the outcrop of the upper bounding surfaces and negative along the outcrop of the lower bounding density surfaces (Figure 4 (a), (b)). On the northern side of the boxes, ocean heat loss drives the upper isopycnal north, increasing Subantarctic mode water volumes. On the southern side of the boxes, ocean heat loss also drives the lower

isopycnal northward, but this makes the volume smaller within the fixed horizontal box. This suggests a dominance of the lateral movement of upper and lower bounding isopycnals rather than vertical movement involving mixed layer depths, which is investigated more fully in Section 4.

After 1.5 years lead time for the central Subantarctic mode water pool the impacts are mainly positive and are either to the north of the upper bounding surface or in between the upper and lower bounding surfaces (Figure 4 (c)), with some small negative impacts mainly to the south of the outcrop of the lower bounding density surface. These impacts follow along the outcropping contours, along the ACC. On lead times of 1.5 years or more for the eastern Subantarctic mode water pool, the impacts are positive locally and to the north of the upper bounding density surface and negative upstream on the southern side of the central region (Figure 4 (d)). Due to the more southerly location of the eastern Subantarctic mode water pool, the outcropping density surfaces are closer together.

At 2.5 years lead time in the central region, there are still mainly positive sensitivities to the north of the upper bounding surface, a mix of impacts in between the upper and lower bounding density surfaces, and negative to the south of the lower bounding density surface (Figure 4 (e)). For the central region, impacts within the northern flow pathway are part of the wider subtropical gyre, and is therefore generally slower than the ACC flow. The forcing in the southern section of the horizontal box is leading to the formation of Subantarctic mode waters that are getting seasonally capped and insulated from the atmosphere to a greater extent than the northern pathway. Therefore, the northern pathway anomalies are more likely to be atmospherically damped. Combining these factors means that longer pathways of sensitivities to the south of each horizontal box are expected. After 2.5 years lead time in the eastern region, the impacts are mainly small when compared to the central region impacts (Figure 4 (f)), and other forcings are similarly important in impacting the volume of the eastern region at this time.

Local and upstream heat fluxes are the dominant contribution to the volume of both Subantarctic mode water pools for up to 3 years in the central pool and up to 2 years in the eastern pool. A longer time series of heat flux impacts may be seen in Figures S6 and S7 for the central and eastern Subantarctic mode water pools respectively.

The heat flux impacts are separated by their position relative to the outcrops of the bounding density surfaces, in order to show the emphasis of flow patterns on the dipole in sensitivities (Figure 5). For the central region, heat flux impacts to the north of 26.7 kg m$^{-3}$, and in between 26.7 - 27.1 kg m$^{-3}$ are mainly positive. The heat flux impacts are also influenced by the seasonal cycle and are much stronger in austral winter (Figure 5 (a)). The negative impacts in the central region are mostly to the south of the 27.1 kg m$^{-3}$ outcrop, and are mainly much smaller than the positive impacts. For the eastern region, the total impacts are smaller than the central region (Figure 5 (b)). Positive impacts are to the north of the upper bounding density surface, 26.9 kg m$^{-3}$. Negative impacts are mainly in between 26.9 - 27.15 kg m$^{-3}$, and to the south of 27.15 kg m$^{-3}$. The northern flow pathways are part of the wider subtropical gyre, and therefore generally slower than the ACC.

The positive and negative heat flux impacts are separated over the bounding density surfaces. Positive impacts are mostly to the north of the upper bounding density surface outcrop for both Subantarctic mode water pools. For the central region, positive impacts are also found in between the upper and lower bounding density surface outcrops. For the eastern region, negative impacts are found in between the upper and lower bounding density surface outcrops. Negative impacts are mostly found to the south of the lower bounding density surface outcrop for both Subantarctic mode water pools. Also, positive impacts

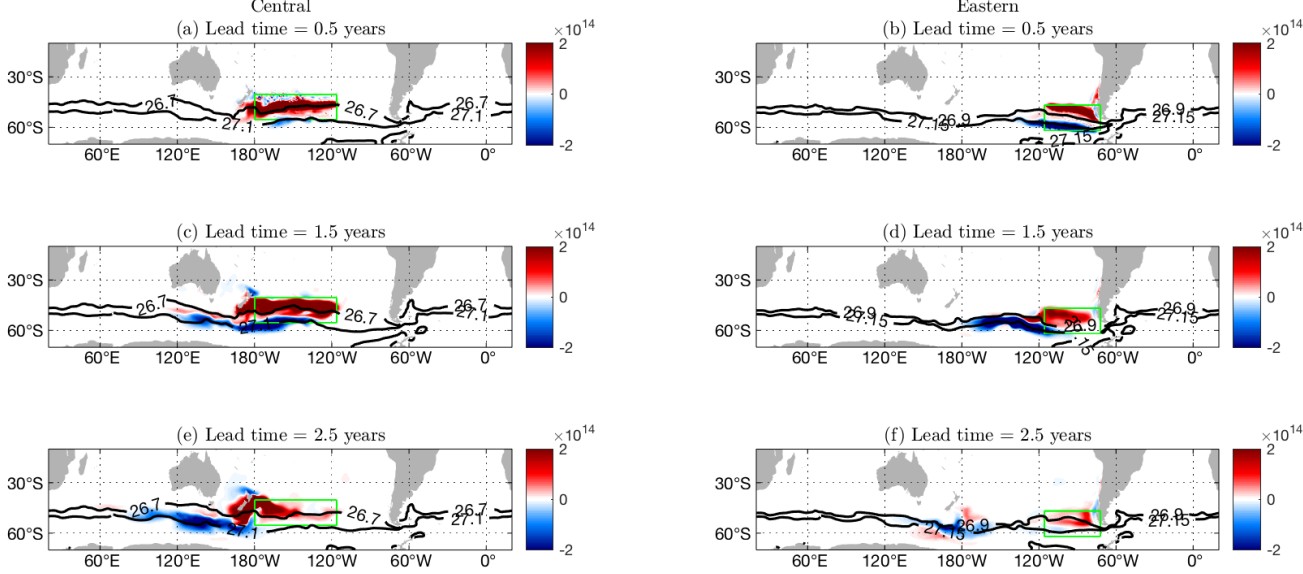

**Figure 4.** (a), (c), (e) Surface heat flux impacts (m$^3$) at the surface for lead times of 0.5, 1.5, 2.5 years. The central adjoint sensitivity target region (green) and the outcrops of the bounding density surfaces (26.7 and 27.1 kg m$^{-3}$, black contours) are included. (b), (d), (f) Surface heat flux impacts (m$^3$) at the surface for lead times of 0.5, 1.5, 2.5 years. The eastern adjoint sensitivity target region (green) and the outcrops of the bounding density surfaces (26.9 and 27.15 kg m$^{-3}$, black contours) are included.

are mostly local, and negative impacts are mostly upstream of the central or eastern adjoint sensitivity region. The local and upstream pattern of the impacts is used to inform the design of the forward perturbation experiments in Section 4.

## 4  Forward perturbation experiments

Forward perturbation experiments are completed in both local and upstream regions of the central and eastern mode water pools to help understand the different physical processes at play. Impacts are mainly positive to the north of the local region, which imply that if there is a cooling with a positive anomaly then there will be an increase in mode water pool volume. Negative impacts are found upstream of both of the control regions, which imply that if there is a warming with a negative anomaly then there will be an increase in mode water pool volume. Hence, the combination of local cooling and upstream warming is the ideal condition to maximise mode water volume. These forward perturbation experiments will help explain how the mode water pool volume increases in both cases.

Forward perturbation experiments are completed in regions with reference to the sensitivity of volume in both the central and eastern pools to heat flux maps. The forward perturbation experiments shown here reference the sensitivity of the volume of the central pool to heat flux only, as the results are mostly similar. Firstly, warming and cooling perturbation experiments are completed in an area local to the central mode water pool. This area is chosen so that it is also an area of positive sensitivity

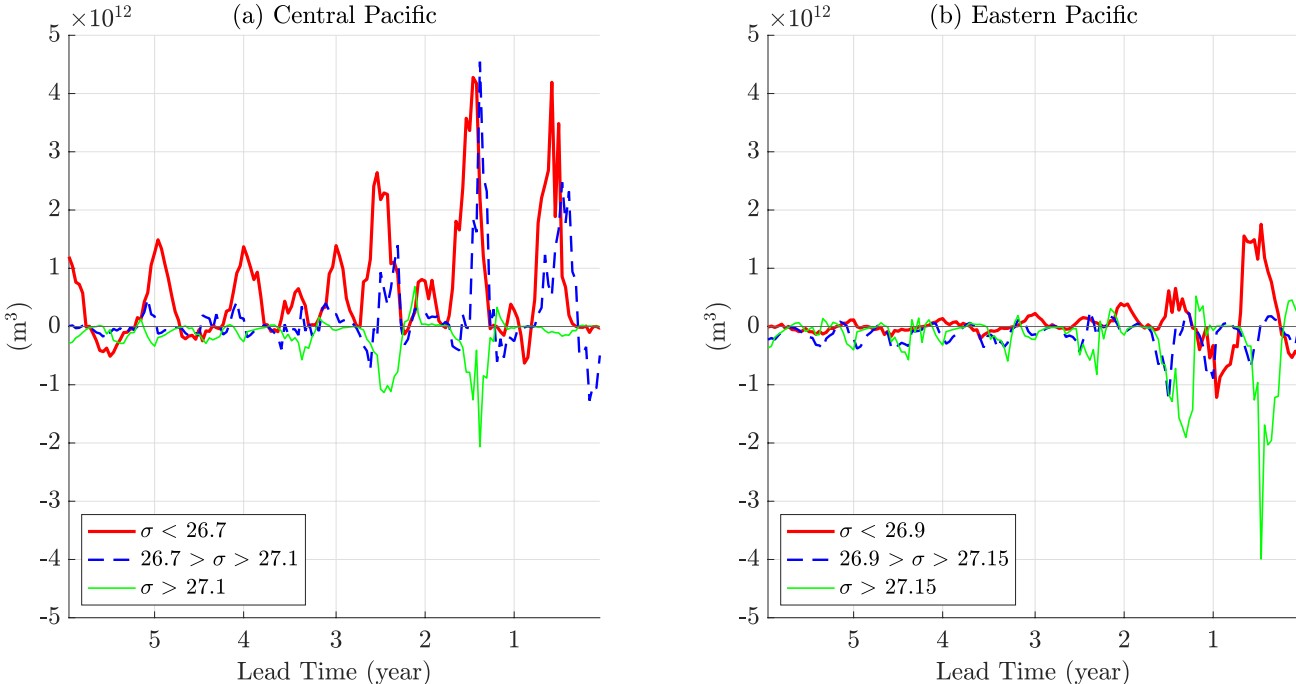

**Figure 5.** (a) Impacts of surface heat flux for the central pool at the surface (anomalies multiplied by sensitivities, m$^3$) averaged over latitude and longitude for the components north of the outcropping density surface 26.7 (red thick), in between the density surface outcrops 26.7 and 27.1 (blue dashed), and to the south of the outcropping density surface 27.1 (green thin) over lead time (year). (b) Impacts of surface heat flux for the eastern pool at the surface (anomalies multiplied by sensitivities, m$^3$) averaged over latitude and longitude for the components north of the outcropping density surface 26.9 (red thick), in between the density surface outcrops 26.9 and 27.15 (blue dashed), and to the south of the outcropping density surface 27.15 (green thin) over lead time (year).

at 1.5 years lead time (Figure 4 (c)). The positive adjoint sensitivities indicate that a warming anomaly will lead to a decrease in volume, and a cooling anomaly will lead to an increase in volume. Next, warming and cooling perturbation experiments are completed in an area upstream of the central mode water pool, located within the ACC. This area is chosen so that it is an area of negative sensitivity at 1.5 years lead time (Figure 4 (c)). Negative adjoint sensitivities indicate that a warming anomaly will lead to an increase in mode water volume, and a cooling anomaly will lead to an decrease in mode water volume. The combination of local cooling and upstream warming increases mode water volume. All four perturbations are of magnitude 100 Wm$^{-2}$, and last for a 3 month time period starting on the 1st June 2016. These local and upstream responses are explored next in more detail.

## 4.1 Local forward perturbation experiments

At 1.5 years lead time the sensitivities of volume to surface heat flux (Figure S3) are both positive and negative, with the positive sensitivities mainly to the north of upper bounding density surface outcrop. The perturbation region is thus defined towards the north of the central box, and the response to this perturbation remains within the northern part of the central box as the mean circulation is weak and the ACC acts as a barrier to the advection of the response. Mode waters are generally formed on the northern edge of the ACC, so this type of perturbation is important for the short term formation of Subantarctic mode waters.

Initially, a local warming perturbation leads to a decrease in neutral density (Figure 6 (a)). The mixed layer depth also begins to shoal and the upper bounding density surface moves further to the south (Figure 6 (c)), creating a smaller volume of mode water in the central pool. The results are consistent with the expectations from the adjoint sensitivity experiments and suggests that nonlinear processes are not important on this timescale.

A cooling perturbation leads to an increase in neutral density when compared to the control experiment (Figure 6 (b)). Cooling leads to the mixed layer deepening and the upper bounding density surface moving further to the north (Figure 6 (d)), and the volume of mode water between the upper and lower bounding density surfaces increases.

The movement of the upper bounding density surfaces does not persist for a long period of time (Figures 6 (e), (f)), although this local process continues to happen over multiple winters. Here it can be seen that local cooling over austral winter will lead to a short term increase in mode water pool volume, as expected. This increase in volume is due to convection, but also the lateral movement of bounding density surfaces.

## 4.2 Upstream forward perturbation experiments

The upstream forward experiments are designed to reveal how upstream warming leads to an increase in the downstream volume of mode waters. At 1.5 years lead time the sensitivities of volume to surface heat flux (Figure S3) are both positive and negative, with the negative sensitivities mainly to the south of the upper bounding density surface and within the ACC, suggesting that advection from upstream source waters may be important.

This warming perturbation is positioned upstream of the central mode water pool within the fronts of the ACC. Warming initially leads to a decrease in neutral density at the site of the warming perturbation (Figure 7 (a)). Warming also leads to a small shoaling of the mixed layer, and laterally moves the outcrop of the lower density surface further to south (Figure 7 (c)). The volume of the central mode water pool is increased as a result. Following the volume response to the perturbation as it is advected along the ACC, the mixed layer depth quickly deepens back to the normal state, but the lateral movement of the lower density surface persists.

The identically positioned cooling perturbation leads to an increase in neutral density, which is of the opposing sign to the warming perturbation (Figure 7 (b)). 3 months after the perturbation, the mixed layer depth deepens and the outcrop of the lower bounding density surface moves laterally the northwards (Figure 7 (d)). The control and perturbation mixed layer depths shoal over the seasonal cycle to the same depths, giving a short lived and local effect on the mixed layer. However,

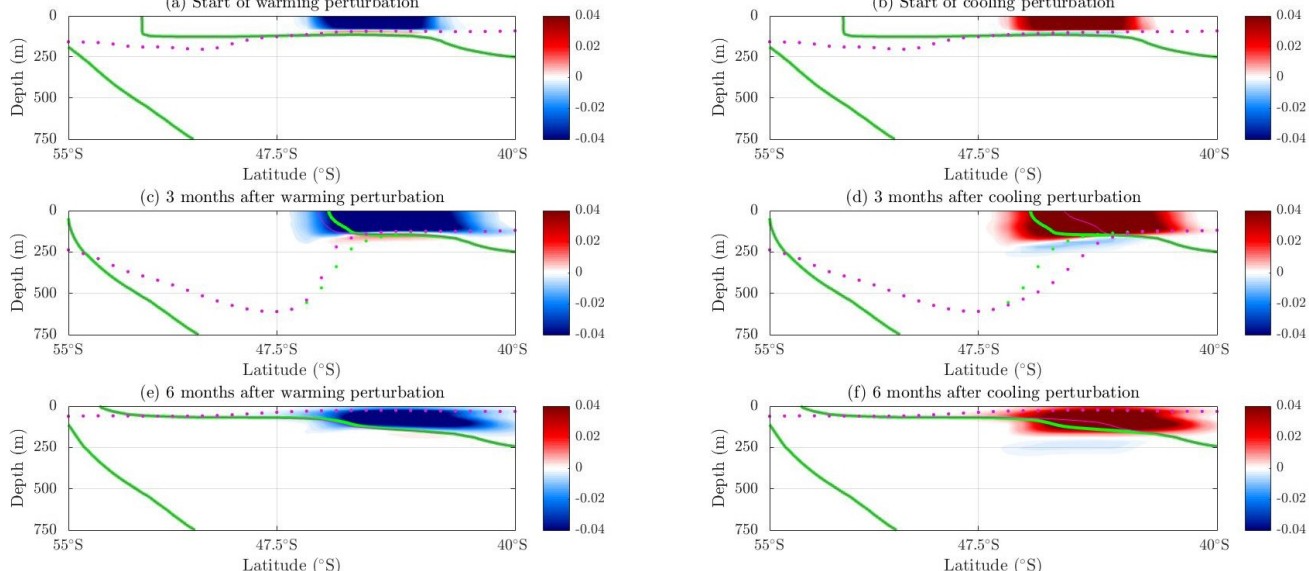

**Figure 6.** Evolution of a negative heat flux perturbation, surface warming, and a positive heat flux perturbation, surface cooling, in a local area of positive sensitivity in the central adjoint experiment for neutral density minus the control neutral density (kg m$^{-3}$) over depth (m) and latitude (degree) at the chosen longitude 148°W (the midpoint of the central pool) at (a), (b) the start of the warming perturbation, (c), (d) 3 months after the perturbation, and (e), (f) 6 months after the perturbation. Magenta dotted lines are perturbation mixed layer depth. Green dotted lines are control mixed layer depth. Thin magenta lines are perturbation neutral density surfaces 26.7 - 27.1 kg m$^{-3}$. Thick green lines are control neutral density surfaces 26.7 - 27.1 kg m$^{-3}$.

the movement of the density surfaces lasts longer and advects along the ACC, with the neutral density changes in the surface ocean.

Hence, through lateral density surface movement, forward perturbations with opposing-sign perturbations lead to similar results with opposing signs, in accord with the adjoint sensitivity and impact analysis.

## 5 Discussion and Conclusions

This study addresses how surface forcing controls Subantarctic mode water volume in the Pacific sector.

Adjoint sensitivities indicate that local cooling leads to an increase in mode water pool volume via mixed layer deepening. In addition, local cooling also leads to the upper density surface outcrop laterally moving to the north, further increasing mode water pool volume. The position of the local cooling is asymmetric, being located over the northern half of the central pool. The central pool is located in a more northerly position than the eastern pool, so this local, northern boundary regime is more important in forcing the central pool.

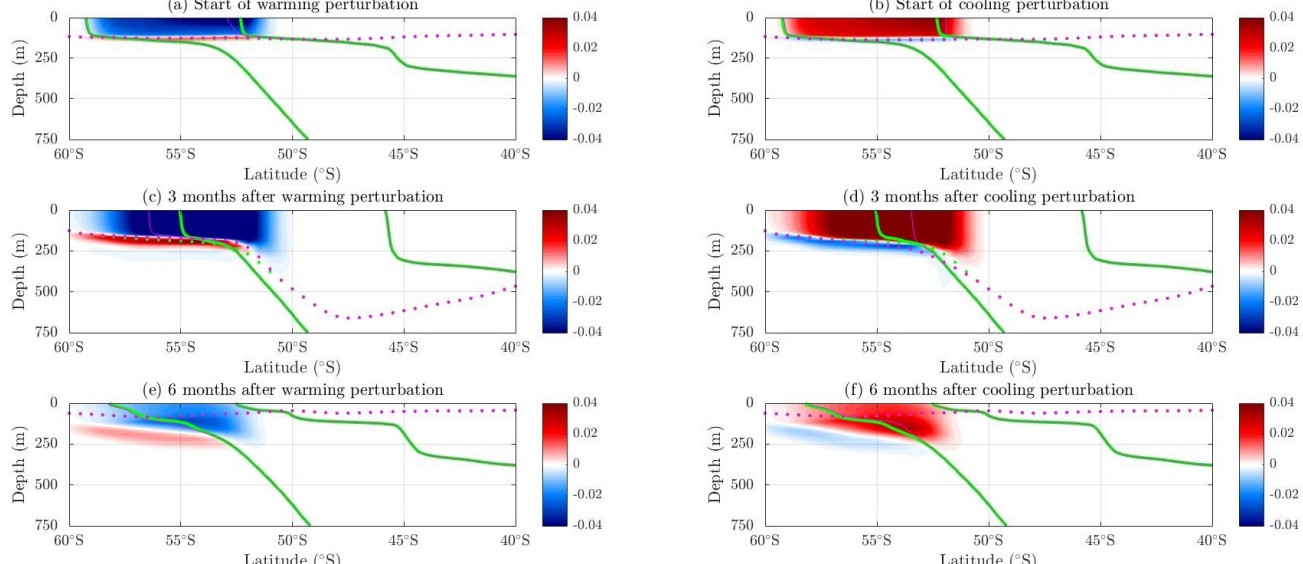

**Figure 7.** Evolution of a negative heat flux perturbation, surface warming, and a positive heat flux perturbation, surface cooling, in an upstream area of negative sensitivity in the central adjoint experiment for neutral density minus the control neutral density (kg m$^{-3}$) over depth (m) and latitude (degree) at the chosen longitude $140°$E (upstream of the Central pool) at (a), (b) the start of the warming perturbation, (c), (d) 3 months after the perturbation, and (e), (f) 6 months after the perturbation. Magenta dotted lines are perturbation mixed layer depth. Green dotted lines are control mixed layer depth. Thin magenta lines are perturbation neutral density surfaces 26.7 - 27.1 kg m$^3$. Thick green lines are control neutral density surfaces 26.7 - 27.1 kg m$^{-3}$.

Adjoint sensitivities also indicate that upstream warming along the ACC leads to an increase in mode water pool volume. The mechanism that leads to this unintuitive result is through the southward, lateral movement of the lower density surface outcrop. The mixed layer shoals from this warming upstream of the SAMW formation region as expected, but this property is not transferred along the ACC and the mixed layer memory in the upstream region is weak after local seasonal shoaling occurs. This southern boundary regime is more important in the eastern mode water pool than the central mode water pool, as upstream impacts are of magnitude $2 \times 10^{12}$ for the eastern but upstream impacts are of magnitude less than $1 \times 10^{12}$ for the central. The eastern pool is located further south than the central pool and more of the eastern pool is controlled by the ACC, which explains the eastern pool impacts being controlled from upstream on lead times longer than 1.5 years. In contrast, the central pool, with its weaker connection with the ACC, is controlled from perturbations more locally. Whilst defining a water mass class by density boundaries will always make its volume sensitive to processes that move those outcropping positions, these bounding surfaces are chosen based on the observed and modelled positions of the deep winter mixed layers associated with SAMW formation. Additionally, this work demonstrates that whilst volumes are most impacted via mixed layer depth changes due to instantaneous forcings consistent with present literature (Meijers et al., 2019; Cerovečki and Meijers, 2021), the associated lateral movement of density surfaces is the most impactful driver of volumes due to upstream forcing.

Optimal conditions for mode water formation involve local, instantaneous cooling and a preconditioning upstream warming one year prior of mode water formation sites. The asymmetric relative positions of the two mode water pools relative to the ACC means that the major atmospheric modes of variability in the Southern Ocean may exert differing impacts on the two pools. The main atmospheric modes in the South Pacific are the ENSO (El Niño Southern Oscillation) and the SAM (Southern Annular Mode). Meijers et al. (2019) considers the instantaneous effects of SAM and ENSO on heat fluxes over the central and eastern mode water pools. Both atmospheric modes lead to a dipole in heat fluxes over the two pools. Positive SAM leads to instantaneous warming over the central box (Meijers et al., 2019), which has been shown here to lead to the lower density surface moving laterally to the south. Over one year this lateral density surface movement is transferred along the ACC and leads to an increase in mode water pool volume in the eastern pool. A positive SAM (or sea level pressure anomaly between the pools) is also associated with cooling over the eastern pool which leads to a local increase in volume. This preconditions the eastern pool to already be thicker before any upstream, advective processes reach the pool. The positive SAM mode affects the southern boundary more than the northern boundary (Meijers et al., 2019), which leads to SAM being the dominant mode over the eastern pool as it is located further south and more within the ACC than the central pool. Recently, SAM has been largely in a positive phase, which reinforces previous work that states that SAMW is getting thicker (Gao et al., 2018). When SAM and ENSO are in phase with each other, ENSO will also reinforce the pattern of SAMW pools becoming thicker. So, this response will be stronger in the South Pacific than in other Southern Ocean mode water formation regions when both ENSO and SAM are in phase with each other.

This study reveals that the southern boundary of the eastern box is influenced significantly by the heat flux and properties of the central box approximate 1 - 1.5 years earlier. This advective timescale is consistent with that suggested by Meijers et al. (2019). Advection may potentially interact with the previously described dipole in mixed layer depth to enhance SAMW formation in the eastern pool. During a dipole year, heat fluxes may be high over the central pool, driving a mixed layer depth shoaling and a dipole deepening of the mixed layer depth in the eastern pool. The warming of the central pool is then advected downstream to the eastern pool the following winter, where that water mass finds a preconditioned 'deep' SAMW layer and acts to enhance the SAMW volume by shifting the isopyncals polewards. In this manner we may expect the eastern pool to be influenced by the central pool, driving deeper mixed layer depths and hence stronger SAMW export. The quantification of this impact is beyond the scope of this study, but is supported by the presence of deeper dipole mixed layers in the eastern than southern pool.

A limitation of this study is the linearity of the adjoint sensitivities. The adjoint sensitivities are linear as only the first derivatives are retained in the calculation. So, any nonlinear processes at play in mode water formation are not captured by the adjoint sensitivities (Forget et al., 2015a). However, the adjoint sensitivities and impacts, are still useful for highlighting important regions and comparing the imprint of different forcings as presented here (Section 3) and in other works (Fukumori et al., 2007a; Pillar et al., 2016; Jones et al., 2020a).

In principle, adjoint calculations similar to the ones shown here could be carried out in the eddy-permitting resolution Southern Ocean State Estimate (Mazloff et al., 2010). However, the higher resolution will likely shorten the timescale on which

the linear adjoint is applicable, potentially limiting the tool's ability to quantify sensitivities to remote forcing. A comparable study using a high-resolution model would be a welcome addition to the literature.

The benefits of using the density-following adjoint feature is that each mode water pool can be better identified using its bounding density surfaces. Previously, adjoint sensitivity experiments (Boland et al., 2021a; Jones et al., 2020a) did not properly capture isopycnal thickness changes which are a defining characteristic of SAMW variability. The density-following feature is used to properly isolate specific water masses only, as defined by their density structure. This prior limitation was particularly an issue in Jones et al. (2016), which attempted to assess changes in exported SAMW properties and struggled to quantify the impacts of changes in properties versus any changes in the density surface positions.

A further caveat of this study is the use of a fixed horizontal box. Over the seasonal cycle the upper and lower bounding density surface outcrops can leave the fixed horizontal box, but this does not occur very often. This movement out of the box leads to a fixed volume being used, at which point the only mechanism for changing the volume is mixed layer depth changes, which is very small. Some aspects may still be further improved to represent changes in the SAMW pool.

In terms of other mode water formations regions, our analysis may be relevant to the South Indian mode water formation region, as central and eastern mode water pools have also been identified (Cerovečki and Meijers, 2021) and as the ACC is important in its formation (Pimm et al., 2024a). The South Indian mode water pool would be a good place to test the ideas put forward here involving preconditioning, role of asymmetry in the pool latitude when compared to the ACC position, and role of different atmospheric modes, such as SAM.

While mode waters in the northern hemisphere are formed in a similar way to mode waters in the Southern Ocean, the northern hemisphere mode waters are generally formed in regions that are influenced by different flow regimes, such as gyres and boundary currents. As a result, these northern hemisphere mode waters may not be as influenced from upstream sources, although the boundary currents in the northern hemisphere may act in a similar way to the ACC and permit the influence of upstream forcing.

In summary, we find that SAMW formation sites are strongly influenced by local instantaneous air-sea heat fluxes. Also, upstream heat fluxes that are subsequently advected into the region play an important role via the lateral movement of density surfaces rather than changes in the mixed layer depth. The exact location of the mixed layer pools relative to the fronts of the ACC and air-sea flux dipoles plays an important role in their response to variability in atmospheric modes, such as SAM or ENSO, and the upstream mode water pool have the potential to significantly influence the downstream pool in subsequent years, leading to a potential net asymmetry in SAMW formation and export.

*Code and data availability.* The data supporting our conclusions can be found at https://doi.org/10.5281/zenodo.12773589 (Pimm, 2024), which includes data from the two separate adjoint experiments in the Central and Eastern South Pacific mode water pools for 6 years lead time. The ECCOv4r4 model setup used in this work is available for download on Github (https://github.com/ECCO-GROUP/ECCO-v4-Configurations/tree/master/ECCOv4%20Release%204) as an instance of the MIT general circulation model (MITgcm, http://mitgcm.org/).

Adjoint code was generated using the TAF software tool, created and maintained by FastOpt GmbH (http://www.fastopt.com/). This work used the ARCHER2 UK National Supercomputing Service (https://www.archer2.ac.uk).

## Appendix A: Appendix A: Density-following adjoint feature

As of checkpoint 68i (Campin et al., 2022), a new feature has been added to MITgcm so that the objective function can be integrated over density surfaces that move in the vertical in time. This process is completed using a sigmoid function, seen here:

```
C ---- If density mask is enabled, use it here ----
            IF ( maskC(i,j,k,bi,bj).EQ.oneRS .AND.
     &            gencost_useDensityMask(kgen) ) THEN
C             - first, calculate the scalar density
              CALL FIND_RHO_SCALAR(
     I                theta(i,j,k,bi,bj),
     I                salt(i,j,k,bi,bj),
     I                gencost_refPressure(kgen),
     O                tmpsig,
     I                myThid )
C             - subtract 1000 to get sigma
              tmpsig = tmpsig - 1000. _d 0
C             - now, tmpmsk is sigmoid times this value
              tmpsig_lower = 0.5 + 0.5*tanh(gencost_tanhScale(kgen)
     &             *(tmpsig-gencost_sigmaLow(kgen)))
              tmpsig_upper = 0.5 - 0.5*tanh(gencost_tanhScale(kgen)
     &             *(tmpsig-gencost_sigmaHigh(kgen)))
C              - update mask value based on the sigmoid function
              tmp_sigmsk = tmpsig_lower*tmpsig_upper
              tmpmsk = tmpmsk*tmp_sigmsk
            ENDIF
C ---- end of density mask (but tmpmsk is used below)
```

Lines 2 and 3 of the code checks if the density following function (gencost_useDensityMask) has been turned on or not by the user in the ECCO namelist file (data.ecco), and also checks that there is a horizontal mask (maskC) present. The mask (maskC) is the chosen horizontal area of interest used in the objective function that is given by the user. Here, kgen is a reference number to make sure that all variables are taken from the same group in the ECCO namelist file, as

there can be many different variable groups to give different adjoint equations. If both of these checks are true then the code continues into the next step (`FIND_RHO_SCALAR`) to calculate the scalar density in lines 5 - 10. If either of the checks fails then this section is skipped and the density-following feature is not used. The function `FIND_RHO_SCALAR` is a preexisting function in the MITgcm code and is used at other times within the model code to calculate scalar density at different time steps. The inputs `theta` and `salt` are found at every latitude, longitude, and depth for the specific time step, which are used in calculating density. The input `gencost_refPressure` is prescribed in `data.ecco` by the user. The output (`tmpsig`) is the temporary sigma value that is used in the sigmoid function to find the upper and lower density surfaces. The final input, `myThid` is the time step that the model is at. In line 12, `tmpsig` has 1000 subtracted so that sigma is in the same form as the user inputs of `gencost_sigmaLow` and `gencost_sigmaHigh`. Lines 14 - 15 and 16 - 17 use a sigmoid function to calculate where the lower (`tmpsig_lower`) and upper (`tmpsig_upper`) density surface is located respectively. Lines 19 - 20 update the vertical mask values (`tmpmsk`) that are then used by the adjoint model equations to calculate the sensitivity of the objective function to the independent variables at this time step.

*Author contributions.* CP, and DJ created and tested the density-following feature; CP performed the modelling experiments; CP analysed the data; CP wrote the manuscript draft; CP, AM, DJ, and RW reviewed and edited the manuscript.

*Competing interests.* There are no competing interests present.

*Acknowledgements.* CP was supported by a UK NERC studentship and by a NERC grant NE/T010657/1 (SARDINE). AM was supported by NERC grant NE/T01069X/1 (SARDINE) and the BIOPOLE National Capability Multicentre Round 2 funding from the Natural Environment Research Council (grant no. NE/W004933/1). DJ was supported by a UK Research and Innovation Future Leaders Fellowship (MR/T020822/1) and also by funding awarded to the Cooperative Institute for Great Lakes Research (CIGLR) through the NOAA Cooperative Agreement with the University of Michigan (NA22OAR4320150). RW was supported by NERC grants NE/T007788/1 and NE/T010657/1 (SARDINE). The authors would like to thank Timothy Smith and David P. Marshall for their insightful reviews.

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
