# Peer review of "Local versus farfield control on South Pacific Subantarctic mode water variability"

_EGUsphere, 2024_

## Referee Comment (RC1)

Review of "Local versus farfield control on South Pacific mode water variability"
By
Ciara Pimm1,4, Andrew J. S. Meijers2, Dani C. Jones2,3, Richard G. Williams1

**Main summary and recommendation**

This study uses the MITgcm and its adjoint modeling capabilities to study the processes that lead to mode water formation in the South Pacific. The authors use adjoint-based sensitivities together with surface flux anomalies to isolate and quantify the impact on mode water formation resulting from surface heat, wind stress, and freshwater flux sources. It is shown that heat fluxes dominate, followed by zonal wind stresses, where impacts typically follow forcing at <1 and ~1.5 year lead times, corroborating previous work. Notably, the work relies on a density based objective function which allows the model to more accurately track mode water pools than previous depth based approaches. Nonlinear forward model perturbation experiments are used to further probe the evolution of heating and cooling.

The paper is well organized and while I have more of a background relevant to the modeling aspects of this paper rather than one relevant to the South Pacific mode water conclusions, I think the study is well motivated and provides useful conclusions to the community. Additionally, I want to additionally congratulate the authors on contributing the density objective function capabilities to the MITgcm code base - this is a highly technical effort which will help to advance the community.

I have reviewed a prior version of this work, and most of my major comments during that review have been addressed. I am therefore recommending minor revisions at this point, given that I only have 1-2 substantive comments at this point.

-Tim Smith

**Conceptual Comments**

- Section 3.1: I totally agree that it is useful to use "impacts" as a way to quantify the sources of variability in SAMW mode water formation (or any other quantity of interest). However, I'm not used to looking at these quantities, so it's hard for me to understand the scales of each of these impact plots (e.g. Figure 3). For example, within the first year local surface heat fluxes generate ~4x10^12 m^3 of SAMW mode water formation. Is this a lot? How does this compare to the seasonal cycle or recent amplitudes of interannual variability? If it's possible to provide any context for these numbers, that would be very helpful.

- Figure 4: I'm still not totally following the interpretation of these plots. Are these generated by accumulating in time from lead_time to 0? Or by simply looking at the "snapshot" of (sensitivity_at_lead_time) * (anomaly_at_lead_time)? Some clarification (or pointing me to the description, in case I missed it) would be very helpful here.

**Small grammatical comments**

- Line 34: At this point in the text there has been no mention of or build up to the use of adjoints, so it feels like it comes out of nowhere. I would recommend moving it to line 65
- Line 163: fromt he -> from the
- Line 164: aand -> and
- Line 272: "both regions local and upstream regions" -> "both local and upstream regions"

---

## Author Response (AR1)

Julian Mak Editor Comments:

In light of the other two referees comments and my reading, I would suggest the authors make a revision incorporating all the comments (mine are given below but I most focused on the writing; some of it overlaps with some referee comments). Given the nature of the comments I think I can make the call on the paper when the revision comes through. – The authors would like to thank the editor for their though review of the manuscript. All the suggested text changes have been completed.

* line 24: space after ACC

* line 34-35: floating sentence? Adjoint hasn't been introduced and reads out of place in my opinion. Probably fits better down near line 65, and brings it closer to the outline, supports that sentence content a bit more. – moved to appropriate section

* somewhere in 2.1 to raise it now, and/or in the conclusion for comparison/outlook: might want to talk about why SOSE was not used (and add related references), presumably because then the non-local contributions would be defined differently (but similar calculations can in principle be performed with SOSE right?) – paragraph added to discussion section

* line 102: suggest "...MITgcm THAT ALLOWS THE objective function TO BE DEFINED AS AN INTEGRAL between two density surfaces THAT..." or similar

* line 108: consider "in the vertical" -> "vertically"

* line 111: suggest comma after "further"

* line 150: I don't think "$J_C$, $J_E$ are the objective functions" is needed, since it's implied in the preceding part already

* line 154 + 155: $\sigma_{uC} = 26.7$ etc. is meant, so I think this could be written out explicitly to help the reader (because I could argue the introduction of the \sigma's are otherwise unnecessary). Similarly with A_C and A_E in the lines below

* line 163: "fromt he" -> "from the"

* line 164: "aand" -> "and"

* line 167: suggest adding comma before "which" to break it up a bit

* line 176: suggest comma after "volume" to break it up a bit

* line 183: me being pedantic, would have "are" -> "may be", and possibly occurrences elsewhere. "are seen in Figure S8" suggests it is an integral part of the article, but that it is in the supplementary material would suggest it is an aside

* line 195: suggest "whereas" -> "By contrast" or similar ("Whereas," as a sentence starter reads a bit funny in this case because of the comma)

* line 196: comma breaks up sentence unnecessarily

* line 206: suggest "dominates" or "is a more dominant contribution" or similar

* paragraph starting line 207: one sentence floating paragraph, consider assimilating it into another paragraph somewhere else (or remove it, since it's not adding much)

* line 214: suggest comma after "in contrast"

* line 216: suggest comma after "lead time"

* line 220: suggest comma after "central region"

* line 227: "dominate" -> "dominant"

* line 228: last part of sentence after the comma reads more convoluted than it needs to in my opinion. I would have done it in active voice ("we focus on the role of the surface heat flux to reveal the underlying mechanism"). To keep it passive voice like the rest of the article I suppose it would be "the present subsection focuses on the role of the surface heat flux to reveal the underlying mechanism" or similar.

line 232: line 233 has "isopycnals" while on this line "isopycnal" is used, so is that intended, and if not, which isopycnal is being referred to here?

line 235: remove "Figure 5" reference, argument being "section 4" details the investigation, but "figure 5" is in support of the investigation (it's just a graph and nothing is investigated there presumably)

* line 239: suggest comma after "pool"

* line 241: suggest comma after "pool"

* line 243: suggest comma after "region"

* line 246: "therefore is" -> "is therefore"

* line 247: "forming" -> "leading to the formation of"

* line 249: reads a bit convoluted, consider "Combining these factors means that longer pathways of sensitivities to the south of each horizontal box are expected"

* line 254: "are shown" -> "may be seen" (for similar reasons in a point above)

* line 258: I think I get what is intended here but there is a redundancy, because "north of 27.1" would cover both cases. Consider re-adjusting words or labelling

* line 262: same as above, "to the south of 26.9" would cover both cases right?

* fig 5 caption: "surfaceheat" -> "surface heat"

* line 283: pretty sure the semi-colon is a typo

* line 286: need a left bracket to close off "Figure 4(c))"

* line 287 + 288: would suggest "The combination of local cooling and upstream warming, again, increases mode water volume." (I don't think it needs ", again," at all to be honest)

* line 289 + 290: suggest "are explored next in more detail"

* line 293 + 294: suggest "The perturbation region is thus defined towards the north of the central box, and ..."

* line 298: suggest "The mixed layer depth also begins..."

* line 300: suggest "The results are consistent with the expectations from the..."

* line 300 + 301: I don't like the "and shows how nonlinear processes are not important on this timescale", because there is no suggestion this is an "if A therefore B" situation. "Show" to me implies a proof and I don't think this is it. Consider simply stating the perturbation results are consistent with the adjoint calculations, or saying the perturbations are CONSISTENT/SUGGESTS that nonlinear processes are not important on this timescale (both being more conservative positions to take, but certainly more defensible).

* line 303 + 304: suggest "...(Figure 6 (d)), AND the volume of..."

* line 317: suggest "The volume of central mode water pool IS INCREASED AS A RESULT."

* line 318: Probably good to be explicit what "control" means here ("control experiment"? presumably not "control variable"? "normal state"?)

* line 322: remove "quite" (subjective, and not entirely necessary adjective)

* line 325: leap-frogging sentence. Consider

"Hence, through lateral density surface movement, forward perturbations with opposing-sign perturbations lead to similar results with opposing signs, in accord with the adjoint sensitivity and impact analysis."

or

"Hence, in accord with the adjoint sensitivity and impact analysis, forward perturbations with opposing-sign perturbations lead to similar results with opposing signs through lateral density surface movement."

or similar.

* line 335: "which" -> "that"

* line 342: suggests commas around "with its weaker connection with the ACC"

* line 344: suggest removing the commas around "and modelled"

* line 346: suggest removing comma after "forcings"

* line 360 + 362: subject associated with "This" can be made more explicit (there are quite a few possibilities here leading to "this" potentially being ambiguous)

* line 374: As David Marshall's comment.

* line 377: would suggest strengthening the "useful" part by referencing to the example given in this present work (reference back to the results around line 300). References to other works would be good too.

* line 379: suggest removing comma after reference

* line 380: remove "So,"

* line 386 + 387: consider "changes, which is very small. Some aspects..."

* line 394 + 395: consider "...regions THAT are influenced by different flow regimes, such as... AS A RESULTS, these... sources, ALTHOUGH..."

* line 399 + 400: there seems to be a missing subject to be tagged with the bit after the semi-colon (not sure what it is supposed to be tagged with)

* line 404, 405 + 406, suggest remove colon, "...(Pimm, 2024), which includes...", and remove "which supports our main conclusions" (because that's implied presumably)

* line 436: "Lines 2 and 3 of the code checkS if..." – I have rewritten parts of this appendix to be more consistent and follow this advice.

* line 437: "...user in the ECCO namelist file (data.ecco)", remove commas surrounding "(maskC)"

* line 440: remove "this"

* line 448: suggest "Lines 14-15 and 16-17 use a ... where the lower (tmpsig_lower) and upper (tmpsid_upper) density surface is located respectively"

* line 450: for consistency ", tmpmask," -> "(tmpmask)"

* bibliography: the Campin et al references have non-standard author names in them, please consider fixing (it's referencing the github usernames, not sure how to fix that automatically though...) – This is the only way that zenodo outputs this reference. I checked the github accounts associated with the non-standard names and they do not list their full name anywhere, so it is too hard to determine who definitely owns the specific github accounts. Therefore, I have left the MITgcm version references to be the same as the previous submission.

* do acknowledge the referees comments etc. if the authors feel they were helpful – Reviewers are thanked in the acknowledgements section.

David Marshall Review:

I enjoyed reading this manuscript. The authors present a series of adjoint sensitivity calculations with the MITgcm to reveal how the volume of Subantarctic mode water is sensitive to remote buoyancy forcing. The surprising conclusion is that the optimal conditions for Subantarctic mode water formation consist of a cooling-heating dipole pattern. This is an interesting result that should be of wide interest. The manuscript is mostly well written and presented, and I therefore happy to recommend it for publication after relatively minor revisions. – The authors would like to the thank David Marshall for their insightful and helpful review of the manuscript. All comments have been considered and the manuscript has been updated to reflect them.

Specific points:

1. Advection of anomalies: In a number of places it is suggested that density anomalies are "advected" (e.g., line 9 of the abstract). While I suspect that advection is dominating the propagation, I am less sure that this has been firmly established and certainly the language should be tightened. In general, density anomalies propagate at the relevant wave speed, which may or may not be dominated by advection. In contrast, the propagation of water mass properties (T-S anomalies on the density surface) and potential vorticity (closely related to the layer thickness in practice) should be dominated by advection. My advice is to carefully check the manuscript to ensure that what is written is accurate when discussing advection, i.e, take care to focus on potential vorticity/layer thickness anomalies rather than density anomalies. – Text has been edited in multiple places to reflect this helpful comment. Now layer thickness is mentioned with reference to advection rather than density.

2. Review of the adjoint method: I was surprised to find the only references in section 2.1 are to papers written by the authors of the present study. While they have made a valuable contribution through the inclusion of a density following feature in the MITgcm adjoint, they are nevertheless building on the shoulders of others who have developed the adjoint machinery within the MITgcm over the past three decades - I think it is fitting and appropriate to give proper acknowledgment to these efforts here. The authors are well aware of this literature. – Thank you for this comment, there has been more information added on the development of adjoint machinery with lots of updated references in the Introduction.

3. Definition of density surfaces: I missed any statement of the definition of the density surfaces following equation (3) and indeed throughout most of the manuscript (an exception being the caption to Figure 6). The authors need to state whether they are using neutral density, potential density (referenced to which depth), or something

else. **– We are using neutral density. The text has been updated in multiple places to reflect this.**

Minor points:

- Line 374: "The adjoint sensitivities are linear due to how the partial derivatives are calculated in the model." This doesn't really make sense - the sensitivities are linear because you have only retained the first derivatives, not because of the way in which the partial derivatives have been calculated. **– The text has been changed to reflect this.**
- Figure 1: The sub-panels are fine to interpret when zoomed in on the screen, but extremely small when viewed at A4 size. I wonder if it you might remove the x- and y-axis labels on all but one of the panels, enabling each panel to be enlarged? Also the labels are in a smaller font than in the remainder of the figures and could be enlarged to help the reader.
- Figure 2: The blue and magenta contour labels are overlapping. It might be better to turn these off the labels for one of the contours since the perturbations are modest. **– This has been corrected, the labels are now only shown for one line colour.**

Typographical issues:

- In the caption to Figure 1, Antarctic Circumpolar Current should be capitalised (and there is a space missing in the third line of the caption). I also noticed that the full name is used three times in the Introduction before the acronym ACC is introduced (and there is a further space missing in line 24). **– ACC acronym has been better defined and used now.**

- General comment: I spotted a small number of minor mis-spellings, so it would be wise to subject the manuscript to a careful spell check. **– Spell check has been completed.**

Timothy Smith Review:

Main summary and recommendation

This study uses the MITgcm and its adjoint modeling capabilities to study the processes that lead to mode water formation in the South Pacific. The authors use adjoint-based sensitivities together with surface flux anomalies to isolate and quantify the impact on mode water formation resulting from surface heat, wind stress, and freshwater flux sources. It is shown that heat fluxes dominate, followed by zonal wind stresses, where impacts typically follow forcing at <1 and ~1.5 year lead times, corroborating previous work. Notably, the work relies on a density based objective function which allows the model to more accurately track mode water pools than previous depth based approaches. Nonlinear forward model perturbation experiments are used to further probe the evolution of heating and cooling.

The paper is well organized and while I have more of a background relevant to the modeling aspects of this paper rather than one relevant to the South Pacific mode water conclusions, I think the study is well motivated and provides useful conclusions to the community. Additionally, I want to additionally congratulate the authors on contributing the density objective function capabilities to the MITgcm code base - this is a highly technical effort which will help to advance the community.

I have reviewed a prior version of this work, and most of my major comments during that review have been addressed. I am therefore recommending minor revisions at this point, given that I only have 1-2 substantive comments at this point.

-Tim Smith – The authors would like to thank Tim Smith for their second detailed and helpful review of this manuscript. Advice from all comments has been incorporated into the updated manuscript.

Conceptual Comments

- Section 3.1: I totally agree that it is useful to use "impacts" as a way to quantify the sources of variability in SAMW mode water formation (or any other quantity of interest). However, I'm not used to looking at these quantities, so it's hard for me to understand the scales of each of these impact plots (e.g. Figure 3). For example, within the first year local surface heat fluxes generate ~4x10^12 m^3 of SAMW mode water formation. Is this a lot? How does this compare to the seasonal cycle or recent amplitudes of interannual variability? If it's possible to provide any context for these numbers, that would be very helpful. – Extra information has been added on how to interpret these impacts in the first 3

- Figure 4: I'm still not totally following the interpretation of these plots. Are these generated by accumulating in time from lead_time to 0? Or by simply looking at the "snapshot" of (sensitivity_at_lead_time) * (anomaly_at_lead_time)? Some clarification (or pointing me to the description, in case I missed it) would be very helpful here. – Line 267, winter 14-day snapshot of impact.

Small grammatical comments – All these grammatical changes have been completed.

● Line 34: At this point in the text there has been no mention of or build up to the use of adjoints, so it feels like it comes out of nowhere. I would recommend moving it to line 65

● Line 163: fromt he -> from the

● Line 164: aand -> and

● Line 272: "both regions local and upstream regions" -> "both local and upstream regions